# Effects of Trehalose Supplementation on Lipid Composition of Rooster Spermatozoa Membranes in a Freeze/Thaw Protocol

**DOI:** 10.3390/ani13061023

**Published:** 2023-03-10

**Authors:** Olga I. Stanishevskaya, Yulia Silyukova, Elena Fedorova, Nikolai Pleshanov, Anton Kurochkin, Vera M. Tereshina, Elena Ianutsevich

**Affiliations:** 1Russian Research Institute of Farm Animal Genetics and Breeding—Branch of the LK Ernst Federal Research Center for Animal Husbandry, Moskovskoe Shosse, 55a, Pushkin, 196625 St. Petersburg, Russia; 2Winogradsky Institute of Microbiology, Research Center of Biotechnology, Russian Academy of Sciences, 119071 Moscow, Russia

**Keywords:** cryopreservation, spermatozoa, rooster, membrane lipids, fatty acids, trehalose

## Abstract

**Simple Summary:**

Sperm cryopreservation is an important part of maintaining the genetic diversity of chicken breeds. However; a significant percentage of spermatozoa lose their viability and motility when frozen/thawed. Cell membranes are the most vulnerable in this process; and their cryoresistance largely depends on their lipid composition. Based on the study of the lipid composition of the plasma membranes of the spermatozoa of roosters of two breeds; a cryoprotective diluent was developed containing biocryoprotective trehalose with a concentration of 9.5 mM; which allows the optimal ratio of cell lipids and the kinetic ability of frozen/thawed spermatozoa (motility) to be maintained at a high level—52.4%.

**Abstract:**

The plasma membrane of spermatozoa plays an important role in the formation and maintenance of many functions of spermatozoa, including during cryopreservation. As a result of chromatographic analysis, the content of lipids and fatty acids in the membranes of spermatozoa of roosters of two breeds was determined under the influence of cryoprotective media containing trehalose LCM-control (0 mM), Treh20 (9.5 mM), and Treh30 (13.4 mM). The use of the cryoprotective diluent Treh20 made it possible to maintain a dynamic balance between the synthesis and degradation of phospholipids and sterols in the plasma membranes of frozen/thawed spermatozoa, close to that of native spermatozoa. This contributed to an increase in the preservation of frozen/thawed spermatozoa membranes from 48.3% to 52.2% in the egg breed and from 30.0% to 35.1% in the meat- and-egg breed. It was also noted that their kinetic apparatus (mobility indicators) remained at the level of 45.6% (egg breed) and 52.4% (meat-and-egg breed). An increase in the concentration of trehalose to 13.4 mM in a cryoprotective diluent for rooster sperm resulted in a decrease in the morphofunctional parameters of frozen/thawed spermatozoa.

## 1. Introduction

The plasma membrane plays an important role in the formation and maintenance of many functions of spermatozoa. There are three main components of the plasma membrane: phospholipids, sterols, and glycolipids. Many studies have determined the special role of phospholipids containing a significant amount of polyunsaturated fatty acids (PUFAs), which provide the necessary degree of membrane fluidity, but at the same time are most susceptible to the harmful effects of reactive oxygen species and peroxidation [1].

When developing new protocols for cryopreservation of avian reproductive cells, it is important to specify the deep mechanisms of formation of the adaptive capabilities of the cell to new conditions. One of the mechanisms of such adaptation may be the modulation of the plasma membrane of the cell that, since it is the plasma membrane, as a cell organelle maintaining the integrity of the cell, is the first to be exposed to external factors. The use of diluents for semen cryopreservation makes it possible to prevent structural changes in the lipid and protein molecules of the plasma membrane of spermatozoa during freezing/thawing of sperm, which is of key importance for the functioning and metabolism of cells.

The decrease in the functionality of spermatozoa in the cryopreservation protocol has been noted by numerous studies. In today’s reproduction technology, there are widely presented methods for determining the decline in the functionality of spermatozoa in terms of their motility, viability, membrane integrity, acrosome integrity, mitochondrial activity, apoptosis status, lipid peroxidation, and fertility [2].

It is known that 70% of the plasma membrane of spermatozoa is composed of lipids and sterols [3]. The matrix of cell plasma membranes includes polar lipids, which consist of hydrophobic and hydrophilic parts [4]. The properties of the spermatozoa membrane and its adaptive capabilities under low-temperature exposure can be predicted by understanding the ratio of its lipid composition. The properties of lipids in the inner and outer layers of the cell membrane can directly correlate with the maintenance and regulation of the composition of the cytosol and the regulation of membrane plasticity.

Lipids are important structural components of the cell plasma membrane and are also a source of energy, being oxidized in mitochondria and peroxisomes [5]. Membrane lipids act as intracellular and extracellular signaling molecules capable of regulating metabolic functions [6]. In the cell nucleus, lipids are involved in the regulation of the expression of genes associated with the metabolism of lipids and carbohydrates [7,8].

The lipids of the inner and outer layers of the plasma membrane differ in composition and perform different functions. Phosphatidylethanolamine (PE) and phosphatidylserine (PS) are characteristic of the inner layer of the membrane and together account for up to 37% of the total composition of phospholipids, with phosphatidylserine present in a smaller amount [9]. PE is the main lipid component of cell membranes in a wide range of organisms and is present exclusively in the inner leaflet of the plasma membrane [10]. Phosphatidylethanolamine is vital for mitochondrial function, as evidenced by defects in oxidative phosphorylation in the absence of PSD1 (phosphatidylserine decarboxylase 1). Lipids also have a key role in the regulation of mitochondrial dynamics in total and protein biogenesis in the outer mitochondrial membrane. Phosphatidylserine (PS), or 1,2-diacyl-sn-glycero-3-phospho-L-serine, is an important anionic phospholipid that imparts significant physical properties to eukaryotic membranes [10].

The outer layer of the plasma membrane of the cell is characterized by a significant amount of phospholipids represented by phosphatidylcholines (PC) and sphingomyelins (SM). Phosphatidylcholines make up 40–50% of the total amount of membrane phospholipids. In this case, the ratio of phospholipids may vary depending on the type of cell or its individual organelles [11].

It is known that trehalose disaccharide is able to protect lipids and proteins of cell plasma membranes from oxidative stress caused by exposure to reactive oxygen species and prevent the oxidation of unsaturated fatty acids. Trehalose protects biological molecules by triggering three mechanisms: replacing water in the cell cytosol, minimizing the formation of ice crystals, and ensuring the chemical stability of biomolecules. Together, these mechanisms make trehalose a highly effective biocomponent of cryoprotective media that ensure the morphological integrity of spermatozoa and their functionality at ultralow temperatures [12,13]

The aim of our study was to determine the composition of lipids in the plasma membranes of rooster spermatozoa (egg and meat-and-egg breeds) and its changes during cryopreservation of male reproductive cells using trehalose disaccharide as a component of cryoprotective media.

## 2. Materials and Methods

### 2.1. Animals

The experiment used roosters—Tsarskoselskaya (TS) (meat-and-egg breed) and Russian White (RW) (egg breed)—kept in individual cages according to the technology of keeping and feeding adopted in the Centre of Collective Usage “Genetic collection of rare and endangered chicken breeds” (RRIFAGB, St. Petersburg, Russia).

### 2.2. Collection, Semen Evaluation, Freezing and Thawing of Rooster Semen

Semen was collected from TS and RW breed roosters (*n* = 10 for each breed) aged 52–56 weeks. Each ejaculate was evaluated individually and selected according to the following criteria: volume using a graduated pipette; spermatozoa concentration using a photometer (IMV Technologies, Bellshill, UK, 2019); total and progressive sperm motility using computer-assisted sperm analysis (CASA) (Motic BA410E, China, 2019, negative contrast, _200; digital input system BASLER acA1300) and software (ArgusSoft, Saint-Petersburg, Russia, 2020). The assessment was carried out in triplicate. The resulting semen from each breed was divided into 4 aliquots. The first aliquot of the native semen was left unchanged, stored at the temperature 5 °C. The other three aliquots were diluted with a separate dilution medium in a 1:1 ratio with a control medium LCM-control [14] and special media containing a disaccharide trehalose (9.5 mM and 13.4 mM) for cryopreservation (Table 1). The diluted semen was cooled for at least 40 min to a temperature of 5 °C. After the samples were cooled, dimethylacetamide (DMA, Sigma Aldrich, St. Louis, MO, USA) was added to them to a final concentration of 6%. The samples were then incubated at 5 °C for 1 min. The prepared semen was frozen in granules by direct dropping of the semen into liquid nitrogen. The position of the Pasteur pipette with sperm was controlled to ensure a temperature range of −15 °C to −20 °C using a digital hand-held thermometer with a sensor (THERM 2420, AHLBORN, Munich, Germany). The pellets were stored for one month. Frozen semen samples were thawed using a metal plate heated to 60 °C (in-house developed equipment, RRIFAGB, 1989).

### 2.3. Viability Assessment of Native and Frozen/Thawed Spermatozoa

The viability of spermatozoa (dilution 1:20) was studied on the basis of histological smears stained with nigrosin-eosin (3% aqueous solution of eosin and 10% nigrosin). The viability of spermatozoa (dilution 1:20) was studied on the basis of histological smears stained with nigrosin-eosin (3% aqueous solution of eosin and 10% nigrosin). Cells were assessed at 1000-fold magnification (Motic BA410E, Hong Kong, China, 2019) by the presence of color. Pink cells were considered dead; unstained (white) cells were considered alive. The results were expressed as percentages of individual categories of spermatozoa (each sample was estimated at 200 cells, which was taken as 100%). The assessment was carried out in triplicate.

### 2.4. Preparation of Semen Samples to Assess the Lipid Composition of Spermatozoa Membranes

To prepare samples for assessing the lipid composition of spermatozoa, semen was collected from 10 males of each breed. Each aliquot (native, frozen/thawed semen with LCM-control medium, Treh20, and Treh30) was processed according to the protocol: Centrifuged for 10 min at 3000 rpm; the supernatant was removed, 0.9% sodium chloride solution was added, thoroughly mixed and centrifuged again under the same parameters. The procedure was repeated three times. At the next stage, the centrifuged sperm were collected in small portions on a nylon filter and precipitated for another 30 min. The prepared samples of centrifuges and supernatants were frozen and stored at −25 °C.

### 2.5. Determination of the Lipid Composition of Native and Frozen/Thawed Spermatozoa

A portion of raw biomass (700–1000 mg) was homogenized in isopropanol, after which lipid extraction was carried out for 30 min at 70 °C and the supernatant liquid was decanted [15]. The residue was extracted twice with isopropanol:chloroform (1:1) and once with a ratio of 1:2 under the same conditions. The combined extract was dehydrated on a rotary evaporator, the residue was dissolved in 9 mL of a mixture of chloroform:methanol (1:1), and 12 mL of 2.5% NaCl was added to remove water-soluble substances. After phase separation, the chloroform layer was taken and dried by passing through anhydrous sodium sulfate, evaporated on a rotary evaporator, and dried to constant weight. The resulting residue was dissolved in a mixture of chloroform:methanol (2:1) and stored at −21 °C.

Separation of polar lipids was performed using 2D TLC on Silica gel 60 glass plates (Merck) in the following systems: chloroform: methanol: toluene: 28% NH_4_OH (65:30:10:6) in the first direction (Vaskovsky system with modification [16]) and chloroform: acetone: methanol: acetic acid: water (50:20:10:10:5) in the second direction (Benning system [17]). A sample (25 μg) of lipids was applied to the plate. The composition of neutral lipids (for the determination of sterols) was analyzed by 1D ascending TLC on Silica gel 60 glass plates (Merck). The solvent system hexane: diethyl ether: acetic acid (77:23:1) was used for separation [18]. A sample (25 μg) of lipids was applied to the plate. The components were visualized by developing chromatograms by spraying with 5% sulfuric acid in ethanol followed by heating for 15 min at 180 °C (total composition of polar lipids). For the identification of lipids, reactions were used with Vaskovsky’s universal reagent with modification of molybdenum blue, Dittmer–Lester reagent, as described [19] (for phospholipids); azur-A (for sulfolipids) [20]; thymol (for glycolipids) [21]; ninhydrin (for lipids containing an amino group); and Dragendorff’s reagent (for choline) as described [18].

Phosphatidylcholine (Sigma, St. Louis, MO, USA) was used as the standard for determining the amount of phospholipids, a mixture of glycoceramides (Larodan, Solna, Stockholm, Sweden) was used for sphingolipids, and ergosterol (Acros Organics, Geel, Belgium) was used for sterols. Quantitative lipid analysis was performed by densitometry using the Dens computer program (Lenchrome, Saint-Petersburg, Russia) in the linear approximation mode using calibration curves based on standard solutions.

### 2.6. Analysis of the Fatty Acid Composition of Membrane Lipids

The fatty acid composition was determined according to the following protocol. The polar fraction of lipids was isolated by one-dimensional TLC. The polar spots of lipids were scraped off, eluted with a mixture of chloroform:methanol (1:1), and the extract was evaporated. Fatty acid methyl esters were obtained by keeping lipids in a 2.5% solution of sulfuric acid in methanol at 70 °C for 2 h [16].

Determination of the composition of lipid fatty acids was carried out on a Kristall 5000.1 ZAO gas-liquid chromatograph (Chromatek, Russia) on an Optima −240, 60 m, 0.25 µm, 0.25 mm capillary column (Macherey-Nagel GmbH&Co, Germany). For chromatography, a temperature program from 130 to 240 °C was used. Identification was carried out using Supelco 37 Component FAME Mix (Supelco, USA).

The degree of unsaturation of phospholipids (DU) was determined by the formula [22]:DU = 1.0 × (% monoene FA)/100 + 2.0 × (% diene FA)/100 + 3.0 × (% triene FA)/100 + 4.0 × (% tetraene FA)/100

### 2.7. Statistical Analysis

For statistical data processing, the software applications Excel 2013 (Microsoft, Redmond, WA, USA) and Statistica 7.0 (StatSoft, Tulsa, OK, USA) were used. Data are presented as mean values standard error (SE) and were considered significant at *p* < 0.05.

## 3. Results

### 3.1. Results of Chromatographic Analysis of the Composition of Membrane Lipids and the Degree of Unsaturation of Fatty Acids in Phospholipids of Rooster Spermatozoa

As a result of chromatographic analysis of the lipids composition in the spermatozoa plasma membranes, interbreed differences were established in the ratio of membrane lipids of native and frozen/thawed spermatozoa (Figure 1a,b; Table 2). In both breeds, major (PE, PA, PS, SM, and ST) lipids ranged from 7.0% to 40.2% of total lipids, while minor (CL, GL, SGL) lipids were 4.0% or less. The composition of the cryoprotective diluent did not affect the lipid composition of the plasma membranes of spermatozoa of native and frozen/thawed rooster sperm (Table 2), but at the same time changed the ratio of these lipids to each other.

The ratio of phospholipids/sterol in spermatozoa membranes, which reflects membrane fluidity, was best preserved in frozen/thawed semen using Treh20 cryoprotective diluent compared to native semen (Figure 2).

The number of phospholipids of the inner (PE + PS) and outer (PC + SM) layers of the plasma membranes of native spermatozoa in both breeds differ little—6.2% and 6.6% (TS) and 7.6% and 7.6% (RW), respectively (Figure 3a,b). It should be noted that the total amount of assessed phospholipids of the inner layer of the plasma membrane of frozen/thawed spermatozoa was increased compared to native sperm in both breeds.

The use of the cryoprotective diluent Treh20 made it possible to maintain the dynamic balance between the phospholipids of the inner and outer membranes of frozen/thawed spermatozoa (TS) at the level of 7.2% and 7.5%, respectively. When using the LCM-control medium, an imbalance of phospholipids of the inner and outer lipid layers of frozen/thawed spermatozoa membranes was noted. The ratio of phospholipids of the outer and inner layers of the plasma membrane of spermatozoa at an increased concentration of trehalose (Treh30) compared with a lower concentration (Treh20) shifted towards the phospholipids of the outer layer. In the Russian White breed, an increase in the content of polar phospholipids of the inner and outer layers of the plasma membrane of spermatozoa was also observed when using the cryoprotective diluent Treh20—8.0% and 10.1%, respectively, compared with native semen.

### 3.2. Determination of Fatty Acid Composition of Membranes of Native and Frozen/Thawed Spermatozoa

A significant variety of fatty acids was determined in the phospholipids of spermatozoa membranes (native and frozen/thawed) (Table 3) in both breeds.

A total of 15 fatty acids have been identified in the lipid composition of the plasma membranes of spermatozoa. In native rooster semen, the predominant fatty acids are the saturated fatty acids palmitic acid (C16:0) and stearic acid (C18:0). The main unsaturated fatty acids in plasma membrane lipids of rooster spermatozoa are docosatetraenoic acid (C22:4), oleic acid (C18:1n9c), and arachidonic acid (C20:4n6) (Table 3).

The use of the cryoprotective diluent Treh20 increased the content of polyunsaturated fatty acids in the membrane lipids of the plasma membranes of spermatozoa of both breeds after freezing/thawing. In the Tsarskoselskaya breed, this value reached 46.5% in native semen and 50.1% in frozen/thawed semen. A similar trend was observed in the Russian White breed in the content of polyunsaturated fatty acids in the lipids of the plasma membranes of spermatozoa during freezing/thawing, and the content of polyunsaturated fatty acids was higher than in the Tsarskoselskaya breed.

In frozen/thawed spermatozoa, the best degree of unsaturation of fatty acids of plasma membrane lipids was determined using the cryoprotective diluent Treh20 in both breeds—1.71 (TS) and 1.81 (RW) (Figure 4).

### 3.3. Results of Evaluation of Sperm Motility and Membrane Integrity in Native and Frozen/Thawed Semen

Evaluation of qualitative parameters of frozen/thawed semen made it possible to determine a cryoprotective diluent with a combined composition of mono- and disaccharides that when used, frozen/thawed semen of roosters has significantly (*p* < 0.05) higher rates of overall motility, progressive motility, and viability of spermatozoa compared to alternative cryoprotective diluents (Table 4).

## 4. Discussion

The profile of the biochemical composition of lipids in the plasma membrane of rooster spermatozoa has been little studied, despite the fact that the lipids of the plasma membrane of spermatozoa are primarily susceptible to the negative influence of cold stress and related factors such as the formation of ROS (reactive oxygen species), lipid peroxidation, a decrease in the level of ATP, etc. ROS-induced damage to spermatozoa in the freeze/thaw protocol is mediated by oxidative ROS attack on polyunsaturated fatty acids that are part of cell membrane phospholipids. This leads to lipid peroxidation and loss of sperm motility and viability [23].

The lipid composition of spermatozoa membranes is quite stable, but under the influence of temperature and the composition of the cryoprotective diluent, the ratio of phospho- and glycolipids, as well as sterol, changes. Sperm plasma membranes of native semen contain a large amount of major phospholipids inherent in the internal layer (phosphatidylethanolamine—22.7% (TS) and 23.9% (RW); phosphatidylserine—18.6% (TS) and 17.6% (RW)) and the external layer (phosphatidylcholine—31.4% (TS) and 37.1% (RW); sphingomyelin—12.9% (TS) and 10.9% (RW)). In addition, one of the major elements of the lipid layer of the rooster spermatozoa plasma membrane for both breeds is sterol, which in our study amounted to 9.7% (TS) and 6.9 % (RW) of the total lipid fraction for native spermatozoa. It is known that cholesterol performs many functions in the biological cycle of the cell: it stabilizes membranes, reduces their permeability, contributes to the stabilization of morphological characteristics and affects the phase transition of the lipid matrix of the membrane at low temperatures, and provides a suitable microenvironment (chemical and/or physical) for membrane proteins [24]. In experiments conducted to study the effect of sterol on the cryoresistance of rooster semen, a negative effect was noted with an increase in the sterol concentration, including an active apoptotic response [25], which can be explained by a decrease in membrane fluidity with a change in lipid concentration in favor of sterol [26,27]. In our studies, the greatest changes in the concentration of sterol in the spermatozoa membrane were observed with the use of the cryoprotective diluent LCM-control. The ratio of the sum of phospholipids and the content of sterol in the cell membrane regulates the degree of fluidity of the spermatozoon membrane, which, according to our assumptions, promotes the transport of saccharides from the external environment into the cell. It is known that membrane transport processes stop when the bilayer viscosity experimentally increases above the threshold level [28].

To ensure the integrity of spermatozoa membranes during cryopreservation, it is necessary to maintain the optimal ratio of phospholipids/sterol. The best cryoprotective effect was achieved when using the Treh20 diluent (trehalose at a concentration of 9.5 mM), which ensured the ratio of phospholipids/sterol in frozen/thawed spermatozoa in both breeds at the level of native semen indicators of 8.6/9.1 (TS) and 12.7/12.6 (RW).

It is known that the level of sphingomyelin, as well as phosphatidylcholine and phosphatidylethanolamine in the plasma membranes of spermatozoa is associated with the percentage of motile spermatozoa and can serve as an indicator of spermatozoa motility and fertility [29,30]. The use of trehalose in cryoprotective diluents (Treh20) made it possible to increase the content of sphingomyelin in the plasma membranes of frozen/thawed spermatozoa to 2.2% (TS) and 2.8% (RW), compared with the control diluent (LCM-control) 2.0% (TS) and 2.3% (RW), which positively affected the total and progressive sperm motility.

The refined composition of minor lipids in spermatozoa membranes (glycolipid, cardiolipin, sulfoglycolipid) was not subject to significant changes either under the influence of the composition of the cryoprotective diluent or under the influence of low-temperature stress.

An analysis of changes in the lipid ratio was carried out taking into account the proven differentiation of phospholipids in the outer and inner layers of membranes [31,32]. The general trend in the quantitative assessment of phospholipids of the outer and inner layers of the plasma membrane of frozen/thawed spermatozoa using the Treh20 medium was an increase in their relative content (% from dry lipids biomass) compared to native sperm, which, in our opinion, was possibly due to the proven antioxidant properties of trehalose [33,34].

The study of the morphological and functional features of the frozen/thawed spermatozoa confirms the positive effect of the combination of mono- and disaccharides in the cryoprotective diluent Treh20 on the indicators of total motility 52.4% (TS)/45.6% (RW) and progressive spermatozoa motility 32.6% (TS)/ 32.4% (RW), respectively, compared with LCM-control total motility 44.5% (TS)/42.4% (RW) and progressive motility 28.3% (TS)/30.0% (RW).

Trehalose, as an antioxidant additive in the Treh20 diluent at a dose of 9.5 mM, had a positive effect on the morphofunctional quality of frozen/thawed spermatozoa, but an increase in the dose to 13.4 mM (Treh30) significantly reduced these parameters compared to Treh20 and LCM-control, which is consistent with the research results of J.-H. Hu et al., 2010 [35]. According to them, an excess amount of trehalose causes destabilizing changes in its own antioxidant systems composed of superoxide dismutase, catalase, reduced glutathione, and glutathione peroxidase [36].

The data obtained agree positively with the results of studies of the effect of sugars under cold stress on the liquid crystalline lamellar phase of cell membrane phospholipids [30] and lipid homeostasis of spermatozoa [24]. The ability of trehalose to protect fatty acids of membrane lipids by forming a special complex with polar groups of macromolecules has been proven [34], significantly reducing their oxidation state and preventing their decomposition [33].

## 5. Conclusions

In our study on two breeds of roosters (egg and meat-and-egg breeds), a dose-dependent effect of trehalose disaccharide, as part of a cryoprotective medium for freezing semen, on the lipid composition of the plasma membrane of spermatozoa, its integrity, and motility of spermatozoa, was revealed. It was found that the optimal concentration of trehalose in the cryoprotective diluent is 9.5 mM (Treh20). The use of the cryoprotective diluent Treh20 made it possible to maintain a dynamic balance between the synthesis and degradation of phospholipids and sterol in the plasma membranes of frozen/thawed spermatozoa, which is close to that of native spermatozoa.

Changes in the composition of cell membrane lipids under the influence of low-temperature stress are a universal natural mechanism for many organisms [37,38]. The spermatozoon, not being an independent organism, at the same time showed a similar response to low-temperature stress, probably due to external resources, i.e., due to the transmembrane transport of trehalose into the cytosol of the spermatozoon.

Apparently, the use of the cryoprotective diluent Treh20 contributed to the maximum preservation of the membrane fluidity of frozen/thawed spermatozoa due to the balance between the amount of phospholipids and sterols and the high content of polyunsaturated fatty acids. This contributed to the preservation of the kinetic apparatus of spermatozoa (motility indicators) at the level of 45.6% (egg breed) and 52.4% (meat-and-egg breed). An increase in the concentration of trehalose to 13.4 mM in a cryoprotective diluent for rooster sperm resulted in a decrease in the morphofunctional parameters of frozen/thawed spermatozoa.

## Figures and Tables

**Figure 1 animals-13-01023-f001:**
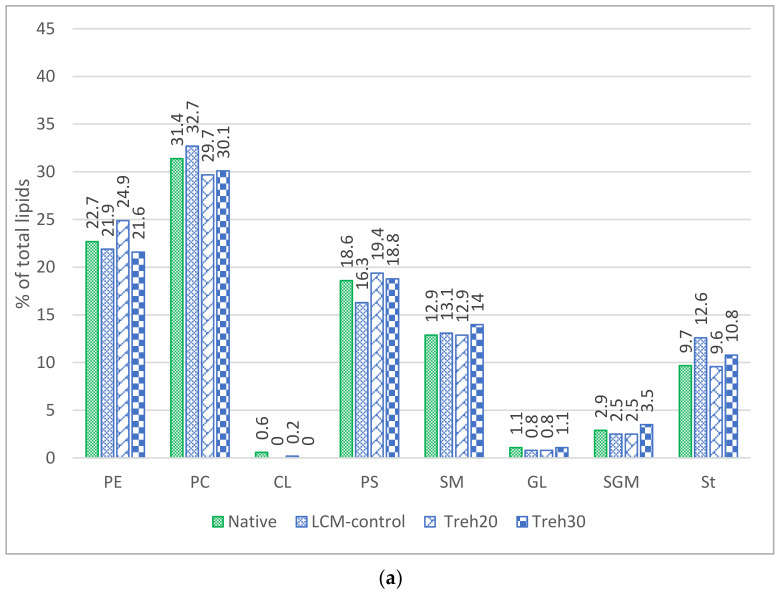
The composition of membrane lipids of spermatozoa of native and frozen/thawed rooster semen depending on the breed and the composition of the cryoprotective diluent: (**a**) native and frozen/thawed TS; (**b**) native and frozen/thawed RW.

**Figure 2 animals-13-01023-f002:**
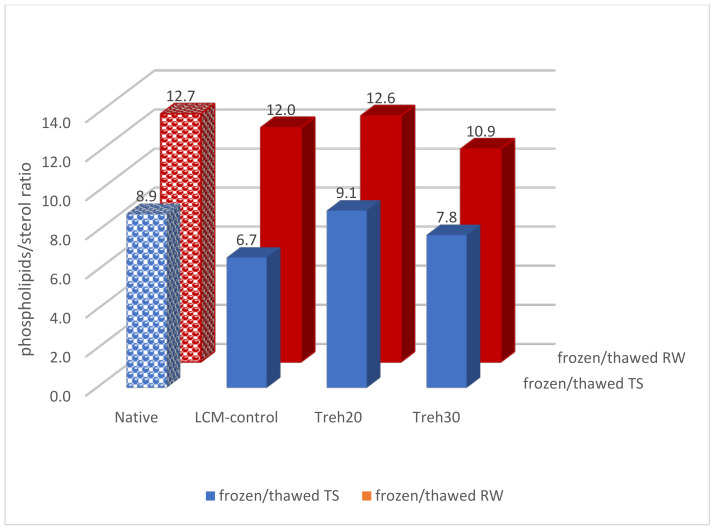
The ratio of the total fraction of phospholipids to the content of sterol in the membranes of spermatozoa of native and frozen/thawed rooster semen depending on the breed and composition of the cryoprotective diluent.

**Figure 3 animals-13-01023-f003:**
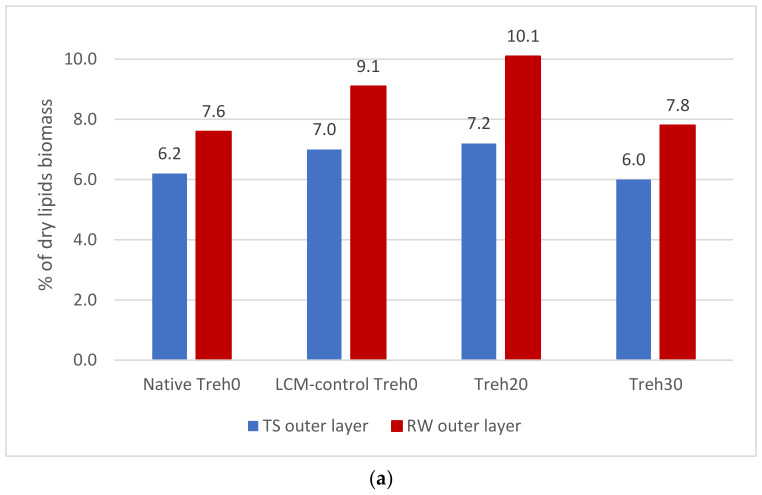
The content of phospholipids of the outer (phosphatidylcholines, sphingomyelins) and inner (phosphatidylethanolamines, phosphatidylserines) layers plasma membrane (% from dry lipids biomass) of native and frozen/thawed spermatozoa (Tsarskoselskaya and Russian White breeds): (**a**) outer layers phospholipids of native and frozen/thawed spermatozoa; (**b**) inner layers phospholipids of native and frozen/thawed spermatozoa. The sum of phospholipids of the inner and outer layers of the plasma membranes of native spermatozoa differs little — 6.2% and 6.6%, respectively.

**Figure 4 animals-13-01023-f004:**
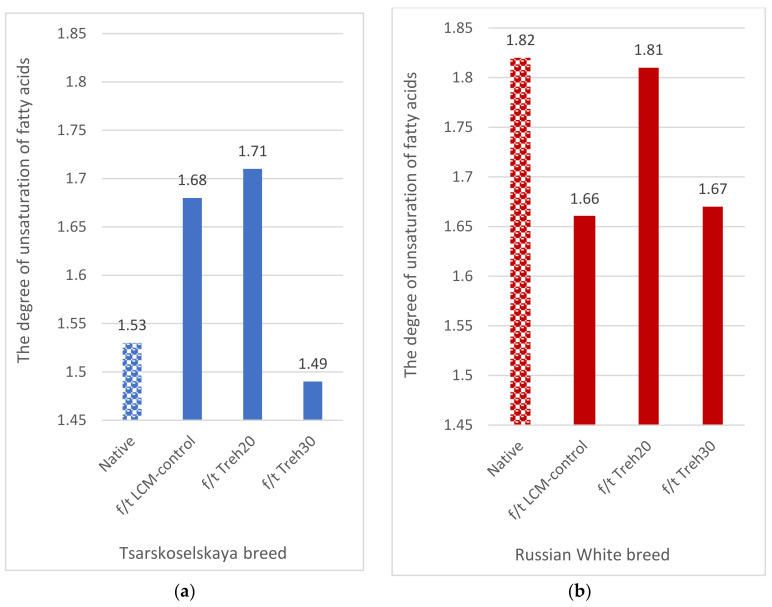
Dynamics of the degree of unsaturation of fatty acids in plasma membranes of spermatozoa of native and frozen/thawed semen depending on the composition of the cryoprotective diluent: (**a**) Tsarskoselskaya breed; (**b**) Russian White breed.

**Table 1 animals-13-01023-t001:** Composition of cryoprotective media for dilution of rooster semen.

Medium Composition	Medium		
	LCM-Control	Treh20	Treh30
Monosodium glutamate	1.92 g (114 mM)	1.92 g (114 mM)	1.92 g (114 mM)
Fructose	0.8 g (44 mM)	0.64 g (36 mM)	0.56 g (31 mM)
Potassium acetate	0.5 g (51 mM)	0.5 g (5 mM)	0.5 g (5 mM)
Polyvinylpyrrolidone	0.3 g (8.3 mM)	0.3 g (8.3 mM)	0.3 g (8.3 mM)
Protamine sulfate	0.032 g (3.27 mM)	0.032 g (3.27 mM)	0.032 g (3.27 mM)
Trehalose	-	0.326 g (9.5 mM)	0.459 g (13.4 mM)
Distilled water	100 mL		
Osmolarity	339 mOsm	344 mOsm	334 mOsm

**Table 2 animals-13-01023-t002:** The lipid composition of spermatozoa membranes of native and frozen/thawed rooster semen depending on the composition of the cryoprotective diluent (dry biomass sample 25 µg).

Lipids, % of Dry Lipids Biomass	Native	TS Freeze/Thawed			Native	RW Freeze/Thawed		
		LCM- Control	Treh20	Treh30		LCM- Control	Treh20	Treh30
phosphatidylethanolamines	3.4	3.4	4.2	3.0	3.8	3.0	4.0	2.8
phosphatidylserines	2.8	2.5	3.3	2.6	2.8	2.6	4.0	3.0
phosphatidylcholines	4.7	5.0	5.0	4.1	5.9	6.8	7.3	5.7
sphingomyelins	1.9	2.0	2.2	1.9	1.7	2.3	2.8	2.1
glycolipids	0.2	0.1	0.1	0.2	0.1	0.2	0.1	0.2
cardiolipins	0.1	0.0	0.0	0.0	0.0	0.0	0.1	0.3
sulfoglycolipids	0.4	0.4	0.4	0.5	0.5	0.7	0.6	0.5
sterols	1.4	1.9	1.6	1.5	1.1	1.2	1.5	1.2

**Table 3 animals-13-01023-t003:** Fatty acid composition of plasma membrane lipids of rooster spermatozoa of native and frozen/thawed semen depending on the composition of the cryoprotective diluent.

Fatty Acids		Tsarskoselskaya				Russian White			
		Native	LCM- Control	Treh20	Treh30	Native	LCM- Control	Treh20	Treh30
C 16:0	Palmitic	20.4	20.4	17.1	21.3	16.4	19.3	16.7	18.8
C 18:0	Stearic	21.4	22.2	21.1	22.7	20.1	21.3	20.2	21.2
C 18:1n9c	Oleic	12.6	13.2	11.7	12.9	11.0	12.1	11.3	11.9
C 18:2n6c	γ-Linoleic	2.6	2.9	2.6	2.0	2.7	3.1	3.0	3.0
C 20:0	Arachidic	1.0	0.7	0.9	1.0	0.7	0.7	0.7	0.8
C 20:1	Arachinoic	3.1	3.2	3.3	3.3	3.5	3.5	3.5	3.5
C 20:2	Eicosadiene	0.6	0.7	0.5	0.4	0.6	0.7	0.6	0.7
C 20:3n6	Eicosatrienoic	1.3	0.9	1.0	0.9	1.0	1.3	1.0	1.1
C 20:4n6	Arachidonic	9.2	10.1	9.6	9.2	10.2	10.1	10.4	9.9
C 22:0	Behenic	0.7	0.3	0.8	0.7	0.2	0.4	0.0	0.4
C 22:1n9	Erucic	0.5	0.3	0.6	0.5	0.4	0.2	0.4	0.4
C 22:2	Docosadiene	2.9	2.8	3.8	3.1	4.4	3.7	4.3	3.9
C 22:4	Docosatetraenoic	20.3	21.1	24.6	20.2	26.4	22.6	25.9	23.1
C 24:0	Lignoceric	0.7	0.3	0.8	0.6	0.4	0.3	0.4	0.5
C 24:1	Nervonic	2.6	0.7	1.7	1.1	1.8	0.6	1.6	0.8
∑ PUFA		46.5	48.5	50.1	45.7	51.7	49.4	52.0	49.4

**Table 4 animals-13-01023-t004:** The morphological and functional features of the native and frozen/thawed rooster semen depending on the composition of the diluent.

Quality Indicators	Native	Frozen/Thawed Sperm		
		LCM-Control	Treh20	Treh30
Tsarskoselskaya breed				
Concentration spermatozoa, billion/mL	3.15 ± 0.13			
Total motility (TM), %	86.3 ± 0.4	44,5 ± 2.5	52.4 ± 1.6 ^a^	32.5 ± 1.5 ^b^
Progressive motility (PM), %	67.5 ± 2.0	28,3 ± 2.5	32.6 ± 1.1 ^a^	21.4 ± 0.4 ^b^
Viability, %	72.6 ± 1.7	30.0 ± 2.0	35.1 ± 0.6 ^a^	26.9 ± 1.5 ^b^
Russian White breed				
Concentration spermatozoa, billion/mL	2.79 ± 0.19			
Total motility (TM), %	86.2 ± 0.0	42.4 ± 2.0	45.6 ± 1.2 ^a^	41.3 ± 0.4 ^b^
Progressive motility (PM), %	66.5 ± 2.5	30.0 ± 1.9 ^a^	32.4 ± 0.5 ^a^	24.4 ± 0.4 ^b^
Viability, %	72.9 ± 5.6	48.3 ± 4.4	52.2 ± 3.2	53.3 ± 2.7

Note: ^a,b^ *p* < 0.05.

## Data Availability

Not applicable.

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
