# Peer review of "Effects of Trehalose Supplementation on Lipid Composition of Rooster Spermatozoa Membranes in a Freeze/Thaw Protocol"

_animals, 2023, doi:10.3390/ani13061023_

Round 1
Reviewer 1 Report (Previous Reviewer 2)
The manuscript has been resubmitted in a greatly improved form that is suitable for publication, within my knowledge of the subject.
Author Response
Dear Reviewer,
thank you for your positive review.
Reviewer 2 Report (Previous Reviewer 1)
I am glad to see that the authors could at least do one more repetition of their key experiment using another rooster breed. Even though this is not enough to test statistical significance, similar effects in both breeds give at least some indication of a potential effect.
It is however important to note that this study shows correlation between lipid composition and survival parameters. No causality between them has been shown. The experiments do not show that survival is better because of lipid composition. They show how lipid composition is under conditions that cause different survival rates. The authors should thus revise their conclusions, in abstract and conclusion sections.
Lines 210-212: How can the lipid ratio change, if the composition is the same?
Lines 240-242: Since only fractions of lipids are analyzed, I assume that the fraction not the total number was increased?
Lines 343-345: Similarly, this is probably relative, not absolute, content.
The last two points open the question, if this should be seen as a positive effect, since increase in one fraction means decrease in another. It should be discussed, what actually decreases and why the authors believe that this has a positive effect.
Author Response
Dear reviewer.
We tried to take into account your comments and make the necessary additions and corrections to the article.
Lines 210-212: How can the lipid ratio change, if the composition is the same?
Answer.
The sentence was changed: The composition of the cryoprotective diluent did not affect the lipid composition of the plasma membranes of the spermatozoa of native and frozen/thawed rooster sperm (Table 2), but at the same time changed the ratio of these lipids to each other.
Lines 240-242: Since only fractions of lipids are analyzed, I assume that the fraction not the total number was increased?
Answer.
The sentence was changed: It should be noted that the total amount of assessed phospholipids of the inner layer of the plasma membrane of frozen/thawed spermatozoa was increased compared to native sperm in both breeds.
Lines 343-345: Similarly, this is probably relative, not absolute, content.
Answer.
The sentence was changed: The general trend in the quantitative assessment of phospholipids of the outer and inner layers of the plasma membrane of frozen/thawed spermatozoa using the Treh20 medium was an increase in their relative content (% from dry lipids biomass) compared to native sperm, which, in our opinion, was possible due to proven antioxidant properties of trehalose [33,34].
The conclusion section has been rewritten
Conclusion
Answer.
The conclusion section has been rewritten: In our study on two breeds of roosters (meat-egg and egg), a dose-dependent effect of trehalose disaccharide as part of a cryoprotective medium for freezing semen on the lipid composition of the plasma membrane of spermatozoa, its integrity and motility of spermatozoa was revealed. It was found that the optimal concentration of trehalose in the cryoprotective diluent is 9.5 mM (Treh20). The use of the cryoprotective diluent Treh20 made it possible to maintain a dynamic balance between the synthesis and degradation of phospholipids and sterol in the plasma membranes of frozen/thawed spermatozoa, which is close to that of native spermatozoa.
Changes in the composition of cell membrane lipids under the influence of low-temperature stress are a universal natural mechanism for many organisms [37,38]. The spermatozoon, not being an independent organism, at the same time showed a similar response to low-temperature stress, probably due to external resources, i.e., due to the transmembrane transport of trehalose into the cytosol of the spermatozoon.
Apparently, the use of the cryoprotective diluent Treh20 contributed to the maximum preservation of the membrane fluidity of frozen/thawed spermatozoa due to the balance between the amount of phospholipids and sterols and the high content of polyunsaturated fatty acids. This contributed to the preservation of the kinetic apparatus of spermatozoa (motility indicators) at the level of 45.6% (egg breed) and 52.4% (meat-and-egg breed). An increase in the concentration of trehalose to 13.4 mM in a cryoprotective diluent for rooster sperm resulted in a decrease in the morphofunctional parameters of frozen/thawed spermatozoa.
This manuscript is a resubmission of an earlier submission. The following is a list of the peer review reports and author responses from that submission.
Round 1
Reviewer 1 Report
Stanishevskaya et al. Present here a study about the lipid content of rooster semen following the treatment with different cryoprotective media depending on the trehalose content. The study design is interesting, and the study seems to show some interesting trends. However, it also has some flaws, which need to be resolved.
1) All measurements on the lipid content have no error. This makes it impossible to judge the significance of the observed changes. If this is data from only one experiment it needs to be repeated.
2) Changing the lipid composition of semen seems to be a quite drastic interference with the functionality of the semen. Optimally, the authors should check the sperm functionality. If this is not possible, they should discuss this issue.
3) Methods are not sufficiently described. E.g. what was the criteria for “morphologically damaged” cells and were they counted as dead or alive? A description of the motility assay seems to misses completely.
4) Minor point: The concentrations given throughout the paper seem to be the concentrations of a stock solution that is diluted, when applied to the cells. It is preferential to give final concentrations.
Author Response
Dear reviewer.
Thank you very much for your attention and reading our article. All the just comments you made have been carefully considered by us, the answers are presented below, as well as in the text of the corrected article marked “Review 1”.
1) All measurements on the lipid content have no error. This makes it impossible to judge the significance of the observed changes. If this is data from only one experiment it needs to be repeated.
Answer.
Sample preparation for chromatographic analysis of the lipid composition of plasma membranes of spermatozoa requires a significant volume of native semen per experiment (the total volume of semen used in this experiment was 210 ml). Given that the volume of rooster ejaculate averages from ~ 0.2 to 1.0 ml, it was not possible to extend the period of semen collection for repeats. In the experiment, we used mixed rooster ejaculates, thus neutralizing individual variability, which provided a reliable model from our point of view.
2) Changing the lipid composition of semen seems to be a quite drastic interference with the functionality of the semen. Optimally, the authors should check the sperm functionality. If this is not possible, they should discuss this issue.
Answer.
In this study, we did not set ourselves the task of determining fertility and limited ourselves to determining the parameters of motility and viability of spermatozoa.
Based on your recommendation, we have made an addition to the Discussion section, where we have shown the relationship between the fertility parameters of frozen/thawed spermatozoa while maintaining the balance of phospholipids/strerols of spermatozoa membranes.
3) Methods are not sufficiently described. E.g. what was the criteria for “morphologically damaged” cells and were they counted as dead or alive? A description of the motility assay seems to misses completely.
Answer.
Thanks for the fair comment. This is our shortcoming.
In paragraph 2.2. and clause 2.3. section Materials and Methods, additions are made.
4) Minor point: The concentrations given throughout the paper seem to be the concentrations of a stock solution that is diluted, when applied to the cells. It is preferential to give final concentrations.
Answer.
These concentrations are given per 100 ml of distilled water (Table 1). The table has been adjusted for better understanding.

Reviewer 2 Report
Highly significant research in the field of cryopreservation of spermatozoa. My comments are in the attached PDF.

Author Response
Dear reviewer.
Thank you very much for your attention and reading our article. All the just comments you made have been carefully considered by us, the answers are presented below, as well as in the text of the corrected article marked “Review 2”.
NOTE (line 23). The concentration of spermatozoa is not mentioned in the text of the methods or the results except in Table 4. Is this the concentration of spermatozoa in the cryosuspension or the initial concentration? Similarly, the volume is only mentioned as 15ml in the pooled sample. This needs more explanatory text.
Answer.
Was added. In paragraph 2.2. “Collection, semen evaluation, freezing and thawing of rooster semen” of the Materials and Methods section has been added: Each ejaculate was evaluated individually and selected according to the following criteria: volume with used graduated pipette; spermatozoa concentration with used photometer (IMV Technologies, Bellshill, UK, 2019); total and progressive sperm motility using CASA (Motic BA410E, China, 2019, negative contrast, _200; digital input system BASLER acA1300) and software (ArgusSoft, Saint-Petersburg, Russia, 2020).
NOTE (line 23). percentage activation, percentage swimming
Answer.
The terms "general and progressive sperm motility" are common terms in reproductive biology. When assessing the parameters of sperm movement by computer-assisted sperm analysis (CASA), this terminology is also used.
(https://doi.org/10.1016/j.psj.2020.10.007, https://doi.org/10.5713/ab.21.0021)
NOTE (line 120). This needs clarification in terms of the constant mM of saccharides, as I read from the table. So the LCM with the semen, was then diluted with "saccharides" in a volume of water ("special medium" as in the text), or as a dry weight to produce the cryosuspension.
Answer.
For a better perception of the material, Table 1 was supplemented.
NOTE (line 283). The results below must be fully described. ie. There was a considerable decline in the percentage of activated, swimming, and viable spermatozoa during freeze/thaw. Nevertheless, the Treh20 treatment provided the highest protection with of spermatozoa with 52% activated, 33% swimming, and 35% viability. Plus more results in text here. NOTE: place approximate symbol in front of values.
Answer.
Corrections have been made and the descriptions of the table have been expanded.
We didn't understand why we should use "approximate symbol" if we were evaluating semen with CASA and the assessment was carried out in triplicate.
NOTE (line 327). This is not clear.
Answer.
The sentence was changed and added. “ The use of the cryoprotective diluent Treh20 contributed to the maximum preservation of the fluidity of frozen/thawed spermatozoa membranes by maintaining the balance between the amount of phospholipids and sterols at the level of native spermatozoa membranes – 9.1 (frozen/thawed semen) and 8.9 (native semen).”
NOTE (line 337). If this occurred it should be clear in the methods and results.
Answer.
The presented conclusions are made on the basis of the results obtained in the section Materials and Methods lines 195-207 and the section Results p.2.2. Determination of fatty acid composition of membranes of native and frozen/thawed 249 spermatozoa lines 249-275.
NOTE (line 352) As the viability, activation, and swimming abilities of spermatozoa after freeze thaw as this final result of the process, there should be more on this. For instance, what are the typical values in other techniques, and importantly how do they relate to fertilisation success?
Answer.
Numerous studies, including those of our scientific group, have determined a positive relationship between an increase in the overall and progressive motility of rooster spermatozoa with their fertility. Therefore, in this study, we did not set ourselves the task of determining fertility and limited ourselves to determining the parameters of sperm motility.
Changes have been made to the text of the article.

Round 2
Reviewer 1 Report
This reviewer understands the constraints associated with sample collection from animals.
However, the assessment of the scientific soundness of this manuscript needs to be based on the presented result. This paper is based on one key experiment, i.e. the measurement of membrane lipid composition after treatment with different media and freeze/thaw cycles. The measured compositions are relatively similar and don’t follow clear trends, i.e. the effects of 20% and 30% trehalose are often in the opposite direction and the same goes for treatments with the same medium with and without freeze/thaw. It is thus absolutely essential to have repeated measurements and appropriate statistical tests to validate any conclusions drawn from this measurement.